# LLBoost: Last Layer Perturbation to Boost Pre-trained Neural Networks

## Abstract

While deep networks have produced state-of-the-art results in several domains from image classification to machine translation, hyper-parameter selection remains a significant computational bottleneck. In order to produce the best possible model, practitioners often search across random seeds or use ensemble methods. As models get larger, any method to improve neural network performance that involves re-training becomes intractable. For example, computing the training accuracy of FixResNext-101 (829 million parameters) on ImageNet takes roughly 1 day when using 1 GPU.

In this work, we present LLBoost, a theoretically-grounded, computationally-efficient method to boost the validation accuracy of pre-trained over-parameterized models without impacting the original training accuracy. LLBoost adjusts the last layer of a neural network by adding a term that is orthogonal to the training feature matrix, which is constructed by applying all layers but the last to the training data. We provide an efficient implementation of LLBoost on the GPU and demonstrate that LLBoost, run using only 1 GPU, improves the test/validation accuracy of pre-trained models on CIFAR10, ImageNet32, and ImageNet. In the over-parameterized linear regression setting, we prove that LLBoost reduces the generalization error of any interpolating solution with high probability without affecting training error.

## 1 Introduction

Over the past decade, deep networks have produced a number of state-of-the-art results including surpassing human performance on the ImageNet classification task (26; 14). However, tuning hyper-parameters to produce the best possible neural network in a given application remains a computationally expensive task. State-of-the-art results often involve selecting the best model using multiple random seeds (14; 27; 12; 28) or ensembling (15), and training even a single model can take several days even when using multiple GPUs. Hence, it is critical to identify computationally efficient approaches to improve the performance of pre-trained models without the need of re-training them.

In this work, we present LLBoost, a theoretically-grounded, computationally-efficient method to boost the validation accuracy of pre-trained, over-parameterized models without impacting the training accuracy. Figure 1 provides an overview of our method as well as the main results. As shown in Figure 1A, our method adjusts the last fully-connected layer of a neural network by selecting the best performing perturbation out of several orthogonal to the training feature matrix, which is constructed by applying all but the last layer to the training data. In Figure 1B, we provide an example showing how our method applied to a model trained under a poorly chosen random seed can boost validation accuracy comparable to that of a model trained under a better seed. Lastly, Figure 1C shows that our method can significantly improve the validation accuracy of pre-trained neural networks on large datasets using a fraction of the training time.

The intuition for our method is based on characterizing the benefit of random initialization in over-parameterized linear regression. In particular, consider a dataset $(X, y) \subset \mathbb{R}^{d \times n} \times \mathbb{R}^{1 \times n}$ with $n < d$ for which there exists $w^* \in \mathbb{R}^{1 \times d}$ such that $y = w^* X$. In order to estimate $w^*$ from the data, we use gradient descent with learning rate $\eta$ and initialization $w^{(0)}$ to minimize the squared loss, i.e. to solve:

$$\arg \min_{w \in \mathbb{R}^d} \frac{1}{2} \|y - wX\|^2.$$

It is well-known (8) that gradient descent converges to the following closed-form solution:

$$w_r = w^{(0)}(I - X(X^T X)^\dagger X^T) + yX^\dagger,$$

where $A^\dagger$ is the pseudo-inverse of $A$ (see Appendix A for a proof). In this work, we prove that when $n < d$, sampling $w^{(0)}$ on a hyper-sphere leads to $w_r$ with lower generalization error than the minimum norm solution $yX^\dagger$ with constant probability. Since the last layer of modern deep networks is usually linear, we apply this result from regression to the last layer of networks to arrive at our method, LLBoost.

We end the introduction with a summary of our main contributions:

1. We present LLBoost, a computationally efficient method for boosting the validation accuracy of pre-trained neural networks without affecting training accuracy. Additionally, we provide an efficient implementation of LLBoost on the GPU.

2. We provide a wealth of empirical evidence that LLBoost improves the validation accuracy of pre-trained neural networks and prove that it does not affect training accuracy for over-parameterized models.

3. We provide evidence that LLBoost yields a computational advantage over random seed search for training neural networks.

4. In the over-parameterized linear regression setting, we prove that LLBoost reduces the test error of any interpolating solution with high probability without affecting training error.

## 2 RELATED WORK

Understanding the benefits of over-parameterization is a recent topic of major interest in machine learning. (2) introduced the double descent curve showing that when increasing model capacity past the interpolation threshold, test error can drop a second time. (20) noted that this phenomenon

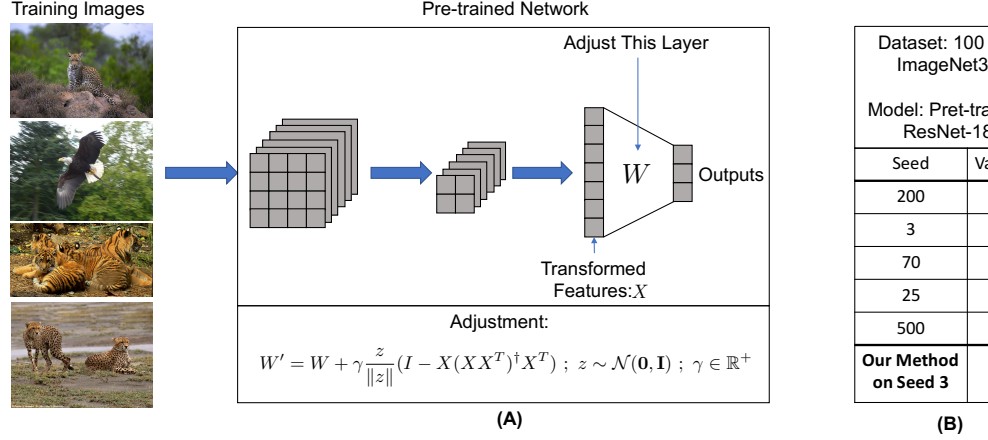

**(A)**

**(B)**

| Dataset: 100 Ex. ImageNet32 | |
|---|---|
| Model: Pret-trained ResNet-18 | |
| Seed | Val Acc. |
| 200 | 80% |
| 3 | 78% |
| 70 | 85% |
| 25 | 80% |
| 500 | 82% |
| **Our Method on Seed 3** | **85%** |

| Dataset | Model | Train/Val. Acc. (Original) | Train/Val. Acc. (Ours) | Training Time | Correction Time |
|---|---|---|---|---|---|
| 100 Ex. ImageNet32 | ResNet-50 | 100% / 83% | 100% / **89%** | 164.07 sec | 0.12 sec |
| CIFAR10 | ResNet-18 | 99.99%/95.05% | 99.99%/**95.25%** | 1.35 hr | 15.36 min |
| ImageNet | FixResNext-101 | 94.92% / 86.26% | 94.92% / **86.34%** | >1 day/epoch | 7.59 hr |

**(C)**

Figure 1: An overview of LLBoost and a demonstration of the benefits of our method in boosting pre-trained models. (A) A breakdown of LLBoost, which improves the performance of pre-trained neural networks by adjusting the last layer of a pre-trained network with a term orthogonal to the training feature matrix, $X$. (B) LLBoost applied to a model trained on a poorly-chosen seed boosts performance to that of a model trained on a well-chosen seed. (C) LLBoost provides a boost to the pre-trained models at a fraction of the computational cost of training.

had been noticed empirically several decades prior to the work of (2). A line of recent work have provided a theoretical analysis of double descent in linear regression (13; 3; 1; 22; 5; 21). In particular, (13; 3) analyzed the generalization error for the minimum norm solution in over-parameterized linear models. These works proved that the minimum norm solution can yield lower generalization error in the over-parameterized regime than in the under-parameterized regime. Our theoretical analysis primarily builds on that of (3), but we analyze the generalization error of random-initialization interpolating solutions instead of the minimum-norm solution.

Various methods, including random seed search and ensembling, are standardly used to improve the performance of machine learning models through training. (4) recommends using 5-10 random seeds if computational power is available, and this is typically done in state-of-the-art models; for example, (14) considered 5 random seeds for ResNet-110 and (12) considered at least 2 random seeds on CIFAR10 (17). (9) rigorously analyzed the impact of random seeds on model performance by considering between 50-400 random seeds for models of varying depth on MNIST (18). Figure 1 from their work demonstrates that random seeds can affect the validation accuracy of models by up to a few percentage points on MNIST. Since this comes at a significant computational price, it is critical to identify methods that can obtain such boosts without having to perform random seed search, which is the topic of this paper.

Another popular approach to improving neural network performance is ensembling, for example via bagging (6) or boosting (10). Recently, (15) presented an approach to ensembling, which involved training a single network and saving the model parameters each time training stopped at a local minimum. (15) built upon (30), which introduced horizontal and vertical voting to ensemble a model across training epochs.

LLBoost can be used in combination with all the above approaches to further boost their performance, since it only adjusts the last layer of a pre-trained model without requiring any training.

## 3 PRELIMINARIES AND METHOD

We now present the preliminaries relevant for formally introducing LLBoost. Since our method is derived from over-parameterized linear regression, we first describe this problem setting.

**Noiseless Regression.** Let $(X, y) \subset \mathbb{R}^{d \times n} \times \mathbb{R}^{1 \times n}$ denote the training dataset where $d$ is the number of dimensions and $n$ is the number of samples. In noiseless linear regression there exists $w^* \in \mathbb{R}^{1 \times d}$ such that $y = w^* X$. In order to estimate $w^*$ from the data, we use gradient descent to minimize the squared loss. Formally, we solve:

$$\arg\min_{w \in \mathbb{R}^d} \frac{1}{2} \|y - wX\|^2 \tag{1}$$

using gradient descent with learning rate $\eta$, which proceeds according to the updates:

$$w^{(t+1)} = w^{(t)} + \eta(y - w^{(t)}X)X^T.$$

The following well-known theorem (8) gives the solution, $w^{(\infty)}$, for the objective (1) given by gradient descent.

**Theorem 1.** *Let $(X, y) \subset \mathbb{R}^{d \times n} \times \mathbb{R}^{1 \times n}$ with $X$ of rank $r < d$, let $\lambda_{max}$ denote the largest eigenvalue of $XX^T$, and let $A^\dagger$ denote the pseudo-inverse of a matrix A. Given initialization $w^{(0)} \in \mathbb{R}^d$, gradient descent used to solve:*

$$\arg\min_{w \in \mathbb{R}^d} \frac{1}{2} \|y - wX\|^2$$

*with learning rate $\eta < \lambda_{max}^{-1}$ converges to $w^{(\infty)} = w^{(0)}(I - X(X^TX)^\dagger X^T) + yX^\dagger$.*

When $w^{(0)} = \mathbf{0}$ (zero initialization), $w^{(\infty)}$ is the minimum $\ell_2$-norm solution for solving the linear system $wX = y$. The following property of $w^{(\infty)}$ is used to demonstrate that LLBoost does not impact training error.

**Lemma 1.** *Let $w^{(\infty)} = w^{(0)}(I - X(X^TX)^\dagger X^T) + yX^\dagger$. Then, $w^{(0)}(I - X(X^TX)^\dagger X^T) \perp yX^\dagger$.*

The proof follows directly by substituting in $X = U \Sigma V^T$ given by the singular value decomposition (SVD). Lemma 1 implies that LLBoost does not affect training predictions since it only ever adds a component orthogonal to the span of the training feature matrix. We primarily focus on the noiseless regression setting, but we note that our theory directly extends to the Gaussian model from (3) and (7).

We now present a result from random matrix theory, which is at the basis of our results. The proof of the following lemma relies on the rotation invariance of the multivariate Gaussian distribution and is presented in Appendix B.

**Lemma 2.** *If $X \in \mathbb{R}^{d \times n}$ with columns $x^{(i)} \sim \mathcal{N}(\mathbf{0}, \mathbf{I}_{d \times d})$, then the singular vectors of $X$ are uniformly distributed on the unit sphere $\mathcal{S}_{d-1}$. In particular, if $x \sim \mathcal{N}(\mathbf{0}, \mathbf{I}_{d \times d})$, then $\frac{x}{\|x\|}$ is uniformly distributed on the unit sphere $\mathcal{S}_{d-1}$.*

**Method Overview.** Given the above preliminaries, we present LLBoost in Algorithm 1. LLBoost takes in the following inputs: (1) the training feature matrix $X$; (2) the validation feature matrix $X_t$; (3) the validation labels $y_t$; (4) the last layer weights $w$; (5) the number of samples to consider $T$; and (6) the radius of the hyper-sphere to sample from $\gamma$. LLBoost begins by constructing the projection matrix $P = I - X(X^T X)^\dagger X^T$. Then, LLBoost samples $z \sim \mathcal{N}(0, I)$, normalizes $z$ and adds the perturbation $\gamma z P$ to $w$. LLBoost repeats this process and returns the perturbation that most improves the validation accuracy.

**Remarks.** As shown in Figure 1A, $X$ and $X_t$ are constructed by applying all but the last layer of the neural network to the training/test data. If $X$ is not full rank (as is the case for neural networks that are not over-parameterized), we can create a low rank $X$ by truncating the lower singular values of $X$. This correction only mildly affects training error as is shown in Section 4. For classification settings, $y_t$ should be converted to 1-hot vectors. When the last layer has a bias, one can simply append a 1 to $w$ and append an extra row of 1's to $X$ and $X_t$. In most of our experiments, we consider up to $T = 100,000$ samples. When selecting $\gamma$, one can utilize a binary search approach starting with $\gamma = \frac{1}{\sqrt{d}}$ and either doubling or halving $\gamma$ depending on whether the validation accuracy increased or decreased respectively. However, as a simple heuristic one can start with $\gamma = \frac{1}{\sqrt{d}}\|w\|$ (as is recommended by Theorem 3). Lastly, when the network has multiple outputs (as in multi-class problems), we let the entries of the weight matrix be i.i.d. standard normal variables and then normalize by the Frobenius norm.

**LLBoost does not impact training error/accuracy.** Since the perturbations constructed by LLBoost are always projected into the space orthogonal to the training features $X$, Lemma 1 implies that LLBoost does not alter the training predictions. This is an important aspect of our algorithm since otherwise, LLBoost could just overfit the test data.

---

**Algorithm 1** LLBoost $(X, X_t, y_t, w, T, \gamma)$

---

**Input:** $X :=$ rank $r$ training feature matrix; $X_t :=$ val. feature matrix; $y_t :=$ val. labels; $w :=$ last layer weights; $T :=$ number of samples; $\gamma :=$ radius of hyper-sphere
**Output:** $w_{best} :=$ adjusted last layer weights

1: $w_{best} = None$
2: $acc_{best} = 0$; $(1, d) = w.shape$; $U, \Sigma, V^T \leftarrow \text{SVD}(X)$
3: $P \leftarrow I_{d \times d} - U I_{r \times r} U^T$
4: **for** $t \leftarrow 1$ to $T$ **do**
5:      $z \leftarrow \mathcal{N}(\mathbf{0}, I_{d \times d})$
6:      $z \leftarrow \frac{z}{\|z\|}$
7:      $w_r \leftarrow \gamma z P + w$
8:      $acc = GetValAcc(X_t, y_t, w_r)$
9:      **if** $acc > acc_{best}$ **then**
10:          $w_{best} = w_r$; $acc_{best} = acc$
11:      **end if**
12: **end for**
13: **return** $w_{best}$

---

| Dataset | Model | Train/Val. Acc. (Original) | Train/Val. Acc. (Ours) | Train/Val. Acc. (Standard Normal) |
|---|---|---|---|---|
| 100 Ex. ImageNet32 | ResNet-18 | 100%/80% | 100%/**84%** | 88%/81% |
| 2600 Ex. ImageNet32 | ResNet-18* | 99.96%/95% | 99.96%/**97%** | 99.96%/95% |
| 100 Ex. ImageNet32 | ResNet-34 | 100%/85% | 100%/**87%** | 74%/82% |
| 2600 Ex. ImageNet32 | ResNet-34* | 99.77%/95% | 99.77%/**98%** | 99.65%/97% |
| 100 Ex. ImageNet32 | ResNet-50 | 100%/83% | 100%/**89%** | 84%/83% |
| 200 Ex. ImageNet32 | ResNet-50 | 100%/87% | 100%/**93%** | 79.5%/87% |
| 500 Ex. ImageNet32 | ResNet-50 | 99.6%/91% | 99.6%/**93%** | 93.2%/92% |
| 800 Ex. ImageNet32 | ResNet-50 | 100%/94% | 100%/**98%** | 97.75%/96% |
| 1000 Ex. ImageNet32 | ResNet-50 | 100%/93% | 100%/**97%** | 98.5%/95% |
| 2000 Ex. ImageNet32 | ResNet-50 | 99.85%/95% | 99.85%/**97%** | 98.8%/97% |
| 2600 Ex. ImageNet32 | ResNet-50* | 99.88%/95% | 99.88%/**99%** | 99.69%/98% |
| CIFAR10 | ResNet-18* | 99.99%/95.05% | 99.99%/**95.25%** | 42.53%/36.46% |
| ImageNet | FixResNext-101* | 94.924%/86.26% | 94.924%/**86.34%** | .57%/.56% |

Figure 2: LLBoost consistently improves the performance of pre-trained neural networks across a variety of experiments on ImageNet32, CIFAR10, and ImageNet without impacting training accuracy. The last column indicates that applying a standard perturbation instead of LLBoost to the last layer of a neural network can significantly impact training and validation accuracy. *'s indicate that the training feature matrix $X$ was full rank, and thus, we used a low rank approximation of $X$ for LLBoost. See Appendix F for a description of how the rank was chosen, and an analysis of how low rank approximations affect train/validation accuracy.

**GPU Implementation.** We can easily remove the loop in Algorithm 1 by using the GPU since the dominating computations are vector-matrix products. A full implementation is provided using PyTorch (25) and NumPy (23; 29) in the footnote below[1]. In practice, computing the validation accuracy for each sample of $w_r$ is the limiting factor in LLBoost. For large datasets, the functions used to compute the SVD and the projection matrix $P$ need to be selected carefully. Note that with respect to SVD, the reduced SVD is sufficient for our algorithm. While for the ImageNet training features from FixResNext-101 the full SVD would require constructing a matrix with roughly $10^{12}$ entries, the reduced SVD only requires $10^8$ entries.

## 4 EMPIRICAL RESULTS

In Figure 2, we provide empirical evidence that LLBoost consistently improves the validation accuracy of state-of-the-art models on CIFAR10, ImageNet32 (24), and ImageNet. For models fine-tuned[2] on ImageNet32 or CIFAR10, we provide all training details in Appendix C. Importantly, as proven by Lemma 1, we note that the training accuracy remains unchanged after applying LLBoost.

**Handling under-parameterized models.** If $n$ denotes the number of training samples, the size of the training feature matrix is $512 \times n$ for ResNets-18 and 34 while it is $2048 \times n$ for ResNet-50 and FixResNext-101 (28). Hence ResNet-18 and 34 are over-parameterized on the training set when $n < 512$, while ResNet-50 and FixResNext-101 are over-parameterized for $n < 2048$. When the training feature matrix, $X$, has full rank, we truncate the lower singular values of $X$ in order to be able to search the space orthogonal to $X$. Formally, if $X = U\Sigma V^T$, we construct $\widehat{X} = U\widehat{\Sigma}V^T$ where we zero out the bottom entries of $\Sigma$. When $\widehat{X}$ has rank $k$, we refer to it as the *rank $k$ approximation* of $X$.

Our experiments are summarized as follows:

1. On ImageNet32, we fine-tune pre-trained ResNets of varying depth on subsets of varying size of two classes (Kit-Fox vs. English-Setter). There are a total of 2600 examples (1300 of each class). We then apply LLBoost to these pre-trained models and use a rank 12

---

[1] https://anonymous.4open.science/r/b11fa900-efbf-45ae-aee8-a264d954cb51/
[2] By fine-tuning, we mean training a pre-trained model on a new dataset. This is discussed further in the transfer learning tutorials provided by PyTorch (25).

approximation for the feature matrix when necessary. This set of experiments demonstrates that LLBoost improves transfer learned models on small datasets.

2. On CIFAR10, we fine-tune a pre-trained ResNet-18. We then apply LLBoost to the pre-trained network. In this setting, there are $50,000$ training examples and so the feature matrix is full rank. To make the model over-parameterized, we instead use the rank 212 approximation. Remarkably, this approximation does not impact the original training accuracy.

3. On ImageNet, we apply LLBoost to pre-trained FixResNext-101. ImageNet contains $1,281,167$ training examples and the feature matrix is again full rank. In this case, we use the rank 1000 approximation, and the original training accuracy decreases from $95.193\%$ to $94.924\%$ when applying the original weights to the rank 1000 approximation.

In all settings, LLBoost improves validation accuracy. In addition, as proven by Lemma 1, there is no impact on training accuracy in the over-parameterized setting. In under-parameterized settings such as FixResNext-101 on ImageNet, our experiments indicate only minor decrease in training accuracy when using a low rank approximation to $X$. In Appendix D, we additionally present: (1) the difference in $\ell_2$/Frobenius norm between the LLBoost weights and the original weights; (2) the run-time of LLBoost; (3) the choice of $\gamma$ used in the experiments. In Appendix F, we discuss how the rank was chosen when approximating full rank feature matrices, and provide an analysis of how low rank approximations affect train/validation accuracy.

**Remarks.** In the last column of Figure 2, we also provide the performance of perturbing the last layer of a neural network with a usual distribution such as the standard normal. Note that such a standard normal perturbation significantly impacts the original training accuracy; in fact for larger datasets such as ImageNet, the training and validation accuracies drop below $1\%$. Hence, without the projection operator, perturbations to the last layer can be devastating to network performance. In Appendix E, we also demonstrate that when including the projection operator but using standard normal initialization (instead of uniform on the hyper-sphere as done in LLBoost) can similarly reduce validation accuracy. While we have thus far concentrated on validation accuracy, in Appendix G we show that in the setting of training, validation, and test splits, LLBoost not only improves validation accuracy, but also improves test accuracy without impacting training accuracy.

**Improvement over Random Seed Search.** We demonstrate in Figure 3 that LLBoost provides a significant computational advantage over random seed search. In particular, in Figure 3 we compare the performance of fine-tuning pre-trained models on ImageNet32 and CIFAR10 using random seeds. Naturally, LLBoost can be applied to boost the performance of all these models. Moreover, Figure 3 illustrates that LLBoost can boost the performance of a model trained using a poor random seed to that of a model trained using a well-selected seed. Since LLBoost only takes a fraction of the training time, these experiments identify LLBoost as a computationally efficient alternative to random seed search.

## 5 THEORETICAL RESULTS

We now present our derivation and analysis of LLBoost for over-parameterized noiseless linear regression. In the noiseless regression setting, we assume that the labels $y \in \mathbb{R}^{1 \times n}$ are generated by the product of $w^* \in \mathbb{R}^{1 \times d}$ and data $X \in \mathbb{R}^{d \times n}$. Let $\widehat{w_r} = w^{(0)}(I - X(X^T X)^\dagger X^T) + yX^\dagger$ and $\widehat{w} = yX^\dagger$ denote interpolating solutions for the linear system $wX = y$ given by gradient descent from initialization $w^{(0)}$ and $\mathbf{0}$ respectively. In this section we establish the following:

1. Generalization bounds on $w_r$.

2. There exist infinitely many $w^{(0)}$ such that $\|\widehat{w_r} - w^*\| < \|\widehat{w} - w^*\|$, i.e. there exist infinitely many random initializations that out-perform the minimum norm solution. In particular, we show that even if $\|w^{(0)} - w^*\|$ is large, $\|\widehat{w_r} - w^*\|$ can be arbitrary close to 0.

3. Sampling $w^{(0)}$ uniformly on the hyper-sphere of appropriate radius leads to $\widehat{w_r}$ out-performing the minimum norm solution with constant probability.

The following proposition (with proof in Appendix H) compares the generalization error of the random initialization solution, $\widehat{w_r}$ and the minimum norm solution, $\widehat{w}$.

| Seed | Val Acc. |
|------|----------|
| 200 | 94% |
| 3 | 94% |
| 70 | 97% |
| 25 | 94% |
| 500 | 95% |
| **LLBoost on Seed 200** | **98%** |

| Seed | Val Acc. |
|------|----------|
| 200 | 95% |
| 3 | 95% |
| 70 | 95% |
| 25 | 97% |
| 500 | 95% |
| **LLBoost on Seed 3** | **97%** |

| Seed | Val Acc. |
|------|----------|
| 200 | 95.05% |
| 3 | 94.98% |
| 70 | 95.06% |
| 25 | 95.14% |
| 500 | 95.1% |
| **LLBoost on Seed 3** | **95.18%** |

**(A)** 800 Ex. ImageNet32 Classes 1, 2 ResNet-50

**(B)** 2000 Ex. ImageNet32 Classes 1, 2 ResNet-50

**(C)** CIFAR10 ResNet-18

Figure 3: LLBoost provides a computational advantage over random seed search. While LLBoost not only improves the performance of every pre-trained model above, our method boosts the performance of a model trained using a poor random seed to at least that of a model trained using a well-selected seed at a fraction of the computational cost.

**Proposition 1.** *Let $X = U\Sigma V^T$ denote the SVD of $X$ and let $\Sigma^\perp = I - \Sigma\Sigma^\dagger$. Then,*

$$(a) \quad \|\widehat{w} - w^*\| = \|w^* U\Sigma^\perp U^T\|,$$

$$(b) \quad \|\widehat{w}_r - w^*\| = \|(w^{(0)} - w^*)U\Sigma^\perp U^T\| \leq \|w^{(0)} - w^*\|.$$

By using Proposition 1 and extending the proof technique of (3), we establish the following generalization bounds for the solution starting from initialization $w^{(0)}$.

**Theorem 2.** *Assuming data $x, x^{(i)} \sim \mathcal{N}(\mathbf{0}, \mathbf{I}_{d \times d})$ for $i \in [n]$, then*

$$(1) \quad \mathbb{E}_{x,X}[(y - \widehat{w}x)^2] = \mathbb{E}_X[\|\widehat{w} - w^*\|^2] = \|w^*\|^2 \left(1 - \frac{n}{d}\right),$$

$$(2) \quad \mathbb{E}_{x,X}[(y - \widehat{w}_r x)^2] = \mathbb{E}_X[\|\widehat{w}_r - w^*\|^2] = \|w^{(0)} - w^*\|^2 \left(1 - \frac{n}{d}\right).$$

The proof is presented in Appendix I. From Theorem 2, the out of sample performance of the solution initialized at $w^{(0)}$ is governed by the distance between $w^{(0)}$ and $w^*$, which matches intuition. While this result is in expectation over the data drawn from an isotropic Gaussian, we now prove the remarkable fact that even if $\|w^{(0)} - w^*\| = c_1$ is large, $\|\widehat{w}_r - w^*\|$ can be any value between $0$ and $c_1$.

**Proposition 2.** *Let $r$ denote the rank of the data matrix $X$ and let $c_1 \geq c_2 \geq 0$. Then there exists $w^{(0)}$ such that $\|w^{(0)} - w^*\| = c_1$ and $\|\widehat{w}_r - w^*\| = c_2$.*

The full proof is presented in Appendix J. Importantly, Proposition 2 provides the following intuition around the benefit of random initialization in over-parameterized models. When $X$ has rank $r$ for $r$ small, the space orthogonal to $X$ is large with dimension $d-r$. There is thus a priori no reason for the true $w^*$ to lie in the span of the training data. Hence, by sampling $w^{(0)}$ to add a term orthogonal to $X$ we can expect to obtain a boost in performance over the minimum norm solution. The following proposition and theorem present a sampling method for $w^{(0)}$, which provably provides a boost over the minimum norm solution with constant probability.

**Proposition 3.** *Let $\mathcal{U}_d$ represent the uniform distribution on the sphere. Assume that the data $x, x_i^{(2)} \sim \mathcal{N}(\mathbf{0}, \mathbf{I}_{d \times d})$ for $i \in [n]$ and that $w^{(0)} \sim \mathcal{U}_d$. Then,*

$$\mathbb{P}_{w^{(0)}}(\mathbb{E}_{x,X}[(y - \widehat{w}_r x)^2] \leq \mathbb{E}_{x,X}[(y - \widehat{w}x)^2]) = \frac{\Gamma\left(\frac{d}{2}\right)}{\sqrt{\pi}\Gamma\left(\frac{d-1}{2}\right)} \int_0^\phi \sin^{d-2}\theta d\theta, \qquad (2)$$

*where $\phi = \cos^{-1}\left(\frac{1}{2\|w^*\|}\right)$ and $\Gamma(x)$ denotes the Gamma function, which satisfies $\Gamma(x) = (x-1)\Gamma(x-1)$ with $\Gamma(1) = 0$ and $\Gamma(\frac{1}{2}) = \sqrt{\pi}$.*

The proof is given in Appendix K. The benefit of Proposition 3 is that we can lower bound the right hand side of Equation (2) by a constant greater than 0. This is computed explicitly in Theorem 3.

**Theorem 3.** *Let $Z_{w^{(0)}} = \mathbb{E}_{x,X}[(y - \widehat{w}x)^2] - \mathbb{E}_{x,X}[(y - \widehat{w}_r x)^2]$ denote the difference in expected generalization error between the minimum $\ell_2$-norm solution and the random initialization solution. If $w_0 \sim \mathcal{U}_d(\gamma)$, $\epsilon \geq 0$, and $\kappa = \frac{\gamma + \frac{\epsilon}{\gamma}}{2\|w^*\|}$, then*

$$\frac{1}{\pi}\cos^{-1}(\kappa) - \frac{\sqrt{d(d-1)}(\sqrt{2(d-4)}+1)\kappa\left(\sqrt{1-\kappa^2}\right)}{2\sqrt{\pi}(d-2)} \leq \mathbb{P}_{w^{(0)}}\left(Z_{w^{(0)}} \geq \epsilon\left(1 - \frac{n}{d}\right)\right) \leq \frac{1}{2}$$

*In particular, if $\|w^*\| = 1$, $\gamma = \frac{1}{\sqrt{d}}$, and $\epsilon = \frac{1}{d}$, then $\kappa = \frac{1}{2\sqrt{d}}$ and*

$$\frac{1}{2} - \frac{1}{2\sqrt{2\pi}} \leq \lim_{d \to \infty} \mathbb{P}_{w^{(0)}}\left(Z_{w^{(0)}} \geq \frac{1}{d}\left(1 - \frac{n}{d}\right)\right) \leq \frac{1}{2} \tag{3}$$

**Remarks.** Theorem 3 is at the core of LLBoost; it implies that the probability of $\widehat{w}_r$ having lower expected generalization error than $\widehat{w}$ is constant. Note that based on Equation (3) this constant is lower bounded by roughly .3 (meaning we get a 30% chance of improvement). Importantly, Theorem 3 can be trivially extended to demonstrate that LLBoost can reduce the generalization error of any interpolating solution with constant probability (provided that the generalization error is not already 0). We also remark that the lower bound in Equation (3) is nearly tight: in practice, this constant is roughly .31. Lastly, note that all of the above theory holds also for the regression setting with multiple outputs by replacing the usual $\ell_2$ norm with the Frobenius norm.

While the full proof of Theorem 3 is presented in Appendix L, we here provide a sketch. The proof provides an approximation for the integral on the right hand side of Equation (2) by using Gautschi's Inequality (11) and integration by parts. Note that while it may seem at first that $\phi$ should be uniformly distributed when $w^{(0)}$ is uniformly distributed on the sphere (and thus the probability should go to 1/2 as $d$ goes to $\infty$), it is in fact not true that $\phi$ is uniformly distributed when $w^{(0)}$ is uniformly distributed, which is why a more precise argument is required.

## 6 DISCUSSION

In this work, we presented LLBoost, a theoretically-grounded, computationally-efficient method to improve the validation accuracy of pre-trained neural networks without impacting training accuracy. Through a variety of experiments on ImageNet32, CIFAR10, and ImageNet, we demonstrated that our method is practically relevant and can be used to boost the performance of state-of-the-art pre-trained models. A standard method to improve the performance of neural networks is random seed search. We showed that LLBoost provides a significant computational advantage over random seed search and can even boost the performance of a model trained on a poorly selected seed to that of a model trained on a well-selected seed. Thus, LLBoost is a computationally efficient alternative to random seed search. Lastly, we provided theoretical footing for our method by showing that LLBoost provably reduces the test error of any interpolating solution in over-parameterized linear regression. An interesting direction of future work is to identify alternate sampling schemes for LLBoost (other than uniform on the hyper-sphere) that provably yield a greater increase in validation accuracy without decreasing training accuracy.

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

APPENDIX

A   GRADIENT DESCENT IN LINEAR REGRESSION

**Theorem 4.** *Let $(X, y) \subset \mathbb{R}^{d \times n} \times \mathbb{R}^n$ with $X$ of rank $r$ and $X = U\Sigma V^T$ its singular value decomposition (SVD). Given an initialization $w^{(0)} = \mathbf{0}$, gradient descent used to solve:*

$$\arg \min_{w \in \mathbb{R}^d} \frac{1}{2} \|y - wX\|^2.$$

*with learning rate $\eta < \frac{1}{\lambda_{max}(XX^T)}$ converges to:*

$$w^{(\infty)} = yV\Sigma^\dagger U^T, \ \ where \ \Sigma^\dagger = \begin{bmatrix} \frac{1}{\sigma_1} & 0 & \ldots 0 & \\ 0 & \frac{1}{\sigma_2} & \ldots 0 & \mathbf{0}_{r \times d-r} \\ 0 & \ldots & \frac{1}{\sigma_r} & \\ & \mathbf{0}_{d-r \times r} & & \mathbf{0}_{d-r \times d-r} \end{bmatrix}.$$

*Proof.* Let $S = XX^T$ and $S' = yX^T$. Then, $w^{(t+1)} = w^{(t)}(I - \eta S) + \eta S'$. Now we directly solve the recurrence relation; namely,

$$w^{(t)} = \eta S'((I - \eta S)^{t-1} + (I - \eta S)^{t-2} + ... + (I - \eta S)^1 + I).$$

Let $X = U\Sigma V^T$ denote the singular value decomposition of $X$ where $\{\sigma_1, \ldots, \sigma_r\}$ are the non-zero entries of $\Sigma$ and $r$ is the rank of $X$. Then, $S = U\Sigma^2 U^T$, and $S' = yV\Sigma U^T$. Thus, we can simplify the recurrence relation:

$$w^{(t)} = \eta S' U((I - \eta\Sigma^2)^{t-1} + (I - \eta\Sigma^2)^{t-2} + ... + (I - \eta\Sigma^2)^1 + I)U^T.$$

Since $(I - \eta\Sigma^2)^{t-1} + (I - \eta\Sigma^2)^{t-2} + ... + (I - \eta\Sigma^2)^1 + I$ is a geometric series, for $\eta < \frac{1}{\sigma_1^2}$, we have:

$$w^{(t)} = \eta S' U\Sigma^+ U^T,$$

$$\Sigma^+ = \begin{bmatrix} \frac{1-(1-\eta\sigma_1^2)^t}{\eta\sigma_1^2} & 0 & \ldots 0 & \\ 0 & \frac{1-(1-\eta\sigma_2^2)^t}{\eta\sigma_2^2} & \ldots 0 & \mathbf{0}_{r \times d-r} \\ 0 & \ldots & \frac{1-(1-\eta\sigma_r^2)^t}{\eta\sigma_r^2} & \\ & \mathbf{0}_{d-r \times r} & & t\mathbf{I}_{d-r \times d-r} \end{bmatrix}.$$

Now substituting in $S' = yV\Sigma U^T$ gives us:

$$w^{(t)} = yV\Sigma^\dagger U^T,$$

$$\Sigma^\dagger = \begin{bmatrix} \frac{1-(1-\eta\sigma_1^2)^t}{\sigma_1} & 0 & \ldots 0 & \\ 0 & \frac{1-(1-\eta\sigma_2^2)^t}{\sigma_2} & \ldots 0 & \mathbf{0}_{r \times d-r} \\ 0 & \ldots & \frac{1-(1-\eta\sigma_r^2)^t}{\sigma_r} & \\ & \mathbf{0}_{d-r \times r} & & \mathbf{0}_{d-r \times d-r} \end{bmatrix}.$$

Lastly, we can take the limit as $t \to \infty$ to conclude that

$$w^{(\infty)} = \lim_{t \to \infty} w^{(t)} = yV\Sigma^\dagger U^T, \ \ where \ \Sigma^\dagger = \begin{bmatrix} \frac{1}{\sigma_1} & 0 & \ldots 0 & \\ 0 & \frac{1}{\sigma_2} & \ldots 0 & \mathbf{0}_{r \times d-r} \\ 0 & \ldots & \frac{1}{\sigma_r} & \\ & \mathbf{0}_{d-r \times r} & & \mathbf{0}_{d-r \times d-r} \end{bmatrix}.$$

$\square$

Note that the proof above can be easily extended to the setting of a random initialization $w^{(0)}$.

## B   Distribution of Singular Vectors of a Random Matrix

*Proof.* We use the rotational invariance of the multivariate isotropic Gaussian. If $A$ is an orthonormal matrix, then we have:

$$x^T I^{-1} x = x^T A^T I^{-1} A x = (Ax)^T I^{-1} (Ax).$$

Now, suppose $A, B$ are both orthonormal matrices, then we have:

$$AXB^T = (A \otimes B) X_v,$$

where $X_v \in \mathbb{R}^{dn}$ is the row-major vectorization of $X$ and $\otimes$ is the Kronecker product. Now, since $A, B$ are orthonormal, we have that $A \otimes B$ is orthonormal. Hence, $AXB^T$ must have the same distribution as $X$, and thus the singular vectors of $AXB^T$ must have the same distribution as those of $X$. Since singular vectors lie on $\mathcal{S}_{d-1}$ and since the distribution is rotation invariant, we conclude that the singular vectors are uniformly distributed on $\mathcal{S}_{d-1}$. □

## C   Training Details

We now describe the training methodology we used to train pre-trained models on ImageNet32 and CIFAR10. The optimizer, initialization, learning rate, and seeds used to train the ResNets in Figure 2 and 3 are presented in Figure 4. Note that all of our models were trained with mean squared error, as discussed in (16). We trained models on ImageNet32 for 150 epochs and on CIFAR10 for 50 epochs. We then saved the model with the highest validation accuracy.

| Dataset | Optimizer, Learning Rate | Initialization | Seed |
|---|---|---|---|
| Classes 1 and 2, ImageNet32 | Adam, 1e-4 | Default Pytorch Initialization | 200 |
| CIFAR10 | Adam, 1e-4 | Default Pytorch Initialization | 200 |

Figure 4: An overview of the optimizer, learning rate, initialization, and seeds used to fine-tune pre-trained models on ImageNet32 and CIFAR10.

For all experiments, we used the PyTorch deep learning library (25). We trained our models on a shared server with 1 Titan Xp and 2 GeForce GTX 1080 Ti's. We only used 1 GPU at a time for training neural networks and applying LLBoost.

## D   Additional Experimental Details

In this section, we provide the following additional details regarding the experiments in Figure 2 and 3:

1. The number of components used in the low-rank approximations for a full rank training feature matrix (Figure 5).

2. The size of the perturbation produced by LLBoost and the values of $\gamma$ used for the models in Figure 2 (Figure 6).

3. A comparison between training time and the time taken for LLBoost to improve the models in Figure 2 (Figure 7).

## E   Performance of projected standard normal perturbations

In Figure 2, we demonstrated that perturbing the last layer without projecting to the space orthogonal to the feature matrix provided a drastic decrease in the training and validation accuracy. In Figure 8, we illustrate the impact of using a perturbation that is randomly sampled from a standard normal and then projected to the space orthogonal to the feature matrix. Again, we see that the validation accuracies can drop significantly for larger datasets in this case. Note that including the projection operator preserves the training accuracy in all cases, as is guaranteed by Lemma 1.

## F   Low Rank Approximations for Feature Matrices

As discussed in Section 4, when the feature matrix, $X$, is full rank, we needed to use a low-rank approximation such that the space orthogonal to $X$. In this section, we discuss our method of

| Dataset | Model | Number of Components |
|---|---|---|
| 2600 Ex. ImageNet-32 | ResNet-18 | 12 |
| 2600 Ex. Imagenet-32 | ResNet-34 | 12 |
| 2600 Ex. Imagenet-32 | ResNet-50 | 1548 |
| CIFAR10 | ResNet-18 | 212 |
| ImageNet | FixResNext-101 | 1000 |

Figure 5: The rank of the approximation used for the training feature matrix, $X$, when $X$ was full rank.

| Dataset | Model | $\gamma$ | Train/Val. Acc. (Original) | Train/Val. Acc. (Ours) | Perturbation |
|---|---|---|---|---|---|
| 100 Ex. ImageNet32 | ResNet-18 | 0.226 | 100%/80% | 100%/84% | 0.21 |
| 2600 Ex. ImageNet32 | ResNet-18* | 11.314 | 99.96%/95% | 99.96%/97% | 11.210 |
| 100 Ex. ImageNet32 | ResNet-34 | 2.263 | 100%/85% | 100%/87% | 2.263 |
| 2600 Ex. ImageNet32 | ResNet-34* | 4.525 | 99.77%/95% | 99.77%/98% | 4.474 |
| 100 Ex. ImageNet32 | ResNet-50 | 0.453 | 100%/83% | 100%/89% | 0.440 |
| 200 Ex. ImageNet32 | ResNet-50 | 0.453 | 100%/87% | 100%/93% | 0.431 |
| 500 Ex. ImageNet32 | ResNet-50 | 4.525 | 99.6%/91% | 99.6%/93% | 3.962 |
| 800 Ex. ImageNet32 | ResNet-50 | 9.051 | 100%/94% | 100%/98% | 7.215 |
| 1000 Ex. ImageNet32 | ResNet-50 | 33.941 | 100%/93% | 100%/97% | 23.784 |
| 2000 Ex. ImageNet32 | ResNet-50 | 45.255 | 99.85%/95% | 99.85%/97% | 6.338 |
| 2600 Ex. ImageNet32 | ResNet-50* | 36.204 | 99.88%/95% | 99.88%/99% | 17.364 |
| ImageNet | FixResNext-101* | 10 | 94.924%/86.26% | 94.924%/86.34% | 0.4428 |
| CIFAR10 | ResNet-18* | 2.0 | 99.99%/95.05% | 99.99%/95.25% | 1.724 |

Figure 6: An extended version of Figure 2 that includes the choice of $\gamma$ considered and the size of the perturbation (in Frobenius norm) produced by LLBoost. $*$'s indicate the use of low-rank approximations for full rank training feature matrices.

choosing the number of components of the SVD to keep for producing the low-rank approximation for $X$. We then present how the number of components selected affects the performance of LLBoost.

In Figure 9, we visualize the normalized singular values of the feature matrix for models from Figure 2. In Figure 9A, we do not use a low-rank approximation as the size of the dataset is already smaller than the number of features. In Figure 9B, the feature matrices are full rank, and so we use a low-rank approximation for the feature matrix with the number of components selected shown red. In particular, we chose a number of components that is well past the elbow in the curve so that there was not a significant drop in training accuracy.

In Figure 10, we demonstrate how the number of components selected for the low-rank approximation affects the validation accuracy of LLBoost. In particular, we observe that using a lower rank approximation generally increases the improvement provided by LLBoost. This matches the intuition provided by Proposition 2: when the space orthogonal to the training feature matrix, $X$, is large, there is no reason to believe that the best linear solution lies in the span of $X$. Hence, sampling the space orthogonal to $X$ yields an improvement. We note that since only a few singular values of $X$ are large, there is no impact to the training accuracy when using a low-rank approximation for $X$ (shown in the second column of the tables in Figure 10).

## G   LLBOOST APPLIED TO TRAIN, VALIDATION, TEST SPLITS

In Figure 2 and Figure 3, we demonstrated that LLBoost improves the validation accuracy of pre-trained models without impacting the training accuracy. To ensure that LLBoost is not overfitting

| Dataset | Model | Training Time | Correction Time (s) |
|---------|-------|---------------|---------------------|
| 100 Ex. ImageNet32 | ResNet-18 | 77.753 sec | 0.116 sec |
| 2600 Ex. ImageNet32 | ResNet-18* | 1020.413 sec | 0.112 sec |
| 100 Ex. ImageNet32 | ResNet-34 | 122.85 sec | 0.111 sec |
| 2600 Ex. ImageNet32 | ResNet-34* | 1397.989 sec | 0.098 sec |
| 100 Ex. ImageNet32 | ResNet-50 | 164.07 sec | 0.113 sec |
| 200 Ex. ImageNet32 | ResNet-50 | 190.473 sec | 0.111 sec |
| 500 Ex. ImageNet32 | ResNet-50 | 407.454 sec | 0.105 sec |
| 800 Ex. ImageNet32 | ResNet-50 | 628.997 sec | 0.137 sec |
| 1000 Ex. ImageNet32 | ResNet-50 | 1054.061 sec | 0.087 sec |
| 2000 Ex. ImageNet32 | ResNet-50 | 1996.991 sec | 0.129 sec |
| 2600 Ex. ImageNet32 | ResNet-50* | 2488.621 sec | 0.11 sec |
| ImageNet | FixResNext-101* | ~1 day/epoch | 7.59 hr |
| CIFAR10 | ResNet-18* | 1.35 hr | 15.36 min |

Figure 7: A comparison between the training time and LLBoost correction time for models from Figure 2. For the ImageNet32 models, the third column represents the time to compute the validation accuracy for $100,000$ samples from LLBoost. For CIFAR10 and ImageNet, the time additionally includes the cost of computing the perturbation for LLBoost. $*$'s indicate the use of low-rank approximations for full rank training feature matrices.

| Dataset | Model | Train/Val. Acc. (Original) | Train/Val. Acc. (Standard Normal @ Perp) |
|---------|-------|----------------------------|-------------------------------------------|
| 100 Ex. ImageNet32 | ResNet-18 | 100%/80% | 100%/76% |
| 2600 Ex. ImageNet32 | ResNet-18* | 99.96%/95% | 99.96%/96% |
| 100 Ex. ImageNet32 | ResNet-34 | 100%/85% | 100%/75% |
| 2600 Ex. ImageNet32 | ResNet-34* | 99.77%/95% | 99.77%/88% |
| 100 Ex. ImageNet32 | ResNet-50 | 100%/83% | 100%/76% |
| 200 Ex. ImageNet32 | ResNet-50 | 100%/87% | 100%/77% |
| 500 Ex. ImageNet32 | ResNet-50 | 99.6%/91% | 99.6%/79% |
| 800 Ex. ImageNet32 | ResNet-50 | 100%/94% | 100%/87% |
| 1000 Ex. ImageNet32 | ResNet-50 | 100%/93% | 100%/96% |
| 2000 Ex. ImageNet32 | ResNet-50 | 99.85%/95% | 99.85%/97% |
| 2600 Ex. ImageNet32 | ResNet-50* | 99.88%/95% | 99.88%/99% |
| ImageNet | FixResNext-101* | 94.924%/86.26% | 94.924/18.54% |
| CIFAR10 | ResNet-18* | 99.99%/95.05% | 99.99%/92.35% |

Figure 8: A demonstration that using samples from a standard normal projected onto the space orthogonal to the training data leads to a decrease in validation accuracy but has no impact on training accuracy. $*$'s indicate the use of low-rank approximations for full rank training feature matrices.

the validation set, we additionally split the validation data into validation and test data and check that LLBoost improves validation and test accuracy without impacting training accuracy[3].

---

[3]For ImageNet32, the validation set size is only 100 examples, and so we split the training set and re-train.

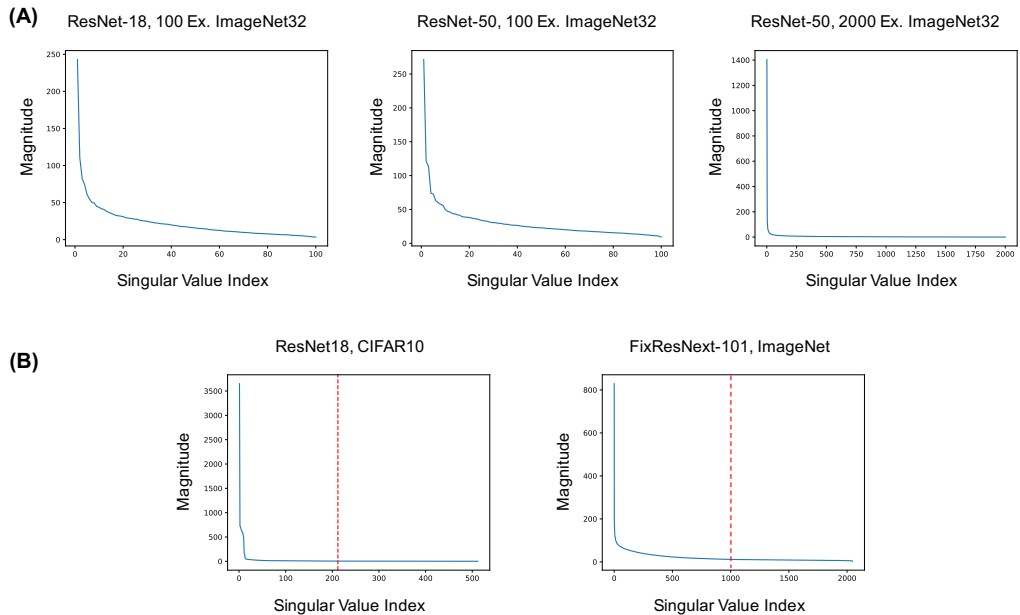

Figure 9: Visualizations of the singular values of training feature matrices for models from 2. (A) The singular values of the training feature matrices for small datasets. (B) The singular values of full rank training feature matrices from large datasets. The red vertical line indicates the size of the approximation used for Figure 2.

| ImageNet32 (2600 Ex.), ResNet-50 | Train Acc. (Original) | Train Acc. (Approx.) | Val. Acc. |
|---|---|---|---|
| Original Feature Matrix | 99.88% | 99.88% | 95% |
| Rank 1548 Approx. | 99.88% | 99.88% | 99% |
| Rank 1648 Approx. | 99.88% | 99.88% | 99% |
| Rank 1748 Approx. | 99.88% | 99.88% | 98% |
| Rank 1848 Approx. | 99.88% | 99.88% | 98% |
| Rank 1898 Approx. | 99.88% | 99.88% | 97% |
| Rank 1948 Approx. | 99.88% | 99.88% | 97% |
| Rank 1973 Approx. | 99.88% | 99.88% | 96% |
| Rank 1998 Approx. | 99.88% | 99.88% | 96% |
| Rank 2023 Approx. | 99.88% | 99.88% | 96% |

| CIFAR10, ResNet-18 | Train Acc. (Original) | Train Acc. (Approx.) | Val. Acc. |
|---|---|---|---|
| Original Feature Matrix | 99.99% | 99.99% | 95.05% |
| Rank 50 Approx. | 99.99% | 99.99% | 95.25% |
| Rank 75 Approx. | 99.99% | 99.99% | 95.25% |
| Rank 100 Approx. | 99.99% | 99.99% | 95.24% |
| Rank 150 Approx. | 99.99% | 99.99% | 95.24% |
| Rank 200 Approx. | 99.99% | 99.99% | 95.23% |
| Rank 212 Approx. | 99.99% | 99.99% | 95.25% |
| Rank 300 Approx. | 99.99% | 99.99% | 95.2% |
| Rank 400 Approx. | 99.99% | 99.99% | 95.19% |
| Rank 500 Approx. | 99.99% | 99.99% | 95.13% |

Figure 10: The impact of using approximations of varying rank for full rank training feature matrices. The first row provides the training accuracy and validation of the original model. The first column is the training accuracy of the model on the original dataset, the second column is the training accuracy on the training data reconstructed from the low-rank approximation, and the third column is the validation accuracy. We see that the validation accuracy generally increases when lowering the rank of the approximation. Since only a few singular values of the training feature matrix are large, there is no impact to the training accuracy when using a low-rank approximation for $X$.

In Figures 11 and 12, we present examples of how LLBoost (which selects the perturbation that improves validation accuracy) improves both validation and test accuracy without impacting training accuracy.

| Dataset | Model | Train/Val./Test Acc.. (Original) | Train/Val./Test Acc. (Ours) |
|---|---|---|---|
| 2600 Ex. ImageNet32 (80/20 split) | ResNet-18 | 99.9%/91.7%/93% | 99.9%/**92.5%/94%** |
| CIFAR10 (20/80 split) | ResNet-18 | 99.99%/95.2%/94.9% | 99.99%/**95.3%/94.93%** |
| CIFAR10 (50/50 split) | ResNet-18 | 99.99%/95.1%/94.82% | 99.99%/**95.16 %/94.84%** |
| CIFAR10 (90/10 split) | ResNet-18 | 99.99%/95.01%/94.50% | 99.99%/**95.04 %/94.60%** |

Figure 11: Using LLBoost to improve validation accuracy also leads to an improvement in test accuracy (i.e. LLBoost does not overfit the validation set). We split the original validation set of CIFAR10 into a validation and test set according to the splits indicated in parentheses. As the validation set of ImageNet32 for 2 classes only has 100 images, we perform an 80/20 train/validation split of the training set, use the 100 validation images as test data, and re-train our models on the smaller training set.

| Dataset | Model | Train/Val./Test Acc. (Original) | Train/Val./Test Acc. (Ours) |
|---|---|---|---|
| 200 Dogs/Cats CIFAR10 | ResNet-18 | 100%/78%/75.50% | 100%/79%/**75.53%** |
| 1000 Dogs/Cats CIFAR10 | ResNet-18 | 100%/84.5%/86.37% | 100%/86%/**86.54%** |
| 2000 Dogs/Cats CIFAR10 | ResNet-18 | 100%/88.65%/89.32% | 100%/89.15%/**89.42%** |

Figure 12: Using LLBoost to improve validation accuracy also leads to an improvement in test accuracy (i.e. LLBoost does not overfit the validation set). In our experiments, we use the same number of examples for training and validation and use the entirety of the remaining examples for testing. For example, in row 1, we use 200 examples for training, 200 for validation and 11600 for testing.

## H    PROOF OF PROPOSTION 1

*Proof.* We first consider $\widehat{w} - w^*$:

$$
\begin{aligned}
\widehat{w} - w^* &= yV\Sigma^{\dagger}U^T - w^* \\
&= w^* XV\Sigma^{\dagger}U^T - w^* \quad (\text{since } y = w^*X) \\
&= w^* U\Sigma\Sigma^{\dagger}U^T - w^*(U\Sigma^{\perp}U^T + U\Sigma^{\perp^{\perp}}U^T) \\
&= w^* U\Sigma^{\perp^{\perp}}U^T - w^*(U\Sigma^{\perp}U^T + U\Sigma^{\perp^{\perp}}U^T) \\
&= -w^* U\Sigma^{\perp}U^T
\end{aligned}
$$

Thus, we have shown (1). Now for (2), we have:

$$
\widehat{w}_r - w^* = w^{(0)}U\Sigma^{\perp}U^T + \widehat{w} - w^* = w^{(0)}U\Sigma^{\perp}U^T - w^*U\Sigma^{\perp}U^T = (w^{(0)} - w^*)U\Sigma^{\perp}U^T.
$$

Hence, (2) follows from (1).   □

## I    PROOF OF THEOREM 2

*Proof.* The proof follows from Lemma 1. Since the columns of $X$ are drawn from $\mathcal{N}(\mathbf{0}, \mathbf{I}_{d \times d})$, Lemma 2 implies that the columns of $U$ are drawn from the uniform distribution on the sphere in $\mathbb{R}^d$. Hence we have that:

$$
\mathbb{E}_X[U\Sigma^{\perp}U^T] = \mathbb{E}_X\left[\sum_{i=n+1}^{d} u_i u_i^T\right] = \sum_{i=n+1}^{d} \mathbb{E}_X[u_i u_i^T] = 1 - \frac{n}{d}.
$$

This implies (1) since:

$$
\mathbb{E}_X[\|\widehat{w} - w^*\|^2 = w^*\mathbb{E}_X[U\Sigma^{\perp}U^T]w^{*T} = \|w^*\|^2 \left(1 - \frac{n}{d}\right).
$$

Similarly, we get (2), which completes the proof.   □

## J  PROOF OF PROPOSITION 2

*Proof.* Let $a^T = w^{(0)} - w^*$. We need to find $a$ such that:

$$(1) \ \|a^T U \Sigma^\perp U^T\|^2 = \sum_{i=r+1}^{d} |\langle a, u_i \rangle|^2 = c_2^2,$$

$$(2) \ a^T a = c_1^2.$$

To do this, we instead first let $a' = c_1 a$ and show that there exists a solution to:

$$(1) \ \|a'^T U \Sigma^\perp U^T\|^2 = \sum_{i=r+1}^{d} |\langle a', u_i \rangle|^2 = \frac{c_2^2}{c_1^2},$$

$$(2) \ a'^T a' = 1.$$

We will show that there is a solution to the above system by using the intermediate value theorem. First, note that the unit sphere is path connected in $\mathbb{R}^d$. Now for $a' = u_{r+1}$, we have $\|a'\| = 1$ and $\|a'^T U \Sigma^\perp U^T\|^2 = 1$. Next, note that for $a' = u_1$, $\|a'\| = 1$ and $|a'^T U \Sigma^\perp U^T|^2 = 0$. Thus, by the intermediate value theorem we conclude that there exists some $a'$ on the unit sphere such that $\|a'^T U \Sigma^\perp U^T\|^2 = \frac{c_2^2}{c_1^2}$, which completes the proof. $\qquad\square$

## K  PROOF OF PROPOSITION 3

*Proof.* Note that we have:

$$\mathbb{P}_{w^{(0)}}\left(\mathbb{E}_{x,X}[(y - \widehat{w}_r x)^2] \le \mathbb{E}_{x,X}[(y - \widehat{w} x)^2]\right)$$
$$\iff \mathbb{P}_{w^{(0)}}\left(\|w^{(0)} - w^*\|^2 \left(1 - \frac{n}{d}\right) \le \|w^*\|^2 \left(1 - \frac{n}{d}\right)\right)$$
$$\iff \mathbb{P}_{w^{(0)}}\left(\langle w^{(0)}, \frac{w^*}{\|w^*\|}\rangle \ge \frac{1}{2\|w^*\|}\right).$$

Since $w^{(0)}$ and $\frac{w^*}{\|w^*\|}$ are unit vectors on $\mathcal{S}_{d-1}$, the desired probability is equivalent to that of the ratio of the area of the spherical cap (19) defined by the co-latitude angle $\phi = \cos^{-1}\left(\frac{1}{2\|w^*\|}\right)$ to the surface area of $\mathcal{S}_{d-1}$, which completes the proof. $\qquad\square$

## L  PROOF OF THEOREM 3

*Proof.* We here present the proof for the case that $\gamma = 1$; however, the proof is easily extendable to the case of arbitrary $\gamma$. The proof relies on the following inequalities, which are commonly used in analysis.

**Proposition 4** (Reduction Formula).

$$\int \sin^d \theta d\theta = -\frac{1}{d} \cos \theta (\sin \theta)^{d-1} + \frac{d-1}{d} \int \sin^{d-2} \theta d\theta$$

**Proposition 5** (Gautschi's Inequality).

$$x^{1-s} < \frac{\Gamma(x+1)}{\Gamma(x+s)} < (x+1)^{1-s} \ ; \ s \in (0,1) \ ; \ x > 0$$

**Corollary 1.** *For $s \in (0,1)$ and $x > 0$:*

$$(1) \ \sqrt{x} < \frac{\Gamma(x+1)}{\Gamma(x+\frac{1}{2})} < \sqrt{x+1},$$

$$(2) \ \frac{1}{\sqrt{x+1}} < \frac{\Gamma(x+\frac{1}{2})}{\Gamma(x+1)} < \frac{1}{\sqrt{x}}.$$

**Proposition 6.**

$$\sum_{i=1}^{k} \frac{1}{\sqrt{i}} \leq \int_0^k \frac{1}{\sqrt{x}} dx = 2\sqrt{k}$$

Let $K = \int_0^\phi (\sin\theta)^{d-2} d\theta$. We will lower bound this integral. For convenience of notation, we will skip writing the limits of integration. By using the reduction formula for the powers of $\int (\sin\theta)^n d\theta$, and assuming $d$ is even for convenience, we have:

$$K = -\frac{1}{d-2}\cos\phi(\sin\phi)^{d-3} - \frac{1}{d-2}\frac{d-3}{d-4}\cos\phi(\sin\phi)^{d-5} - \ldots - \frac{(d-3)!!}{(d-2)!!}\cos\phi\sin\phi + \frac{\Gamma(\frac{d-1}{2})}{\sqrt{\pi}\Gamma(\frac{d}{2})}\phi$$

$$= -\frac{1}{d-2}\cos\phi\sin\phi\frac{\Gamma(\frac{d-1}{2})}{\sqrt{\pi}\Gamma(\frac{d-2}{2})}\left[\frac{\sqrt{\pi}\Gamma(\frac{d-2}{2})}{\Gamma(\frac{d-1}{2})}(\sin\phi)^{d-4} + \frac{\sqrt{\pi}\Gamma(\frac{d-4}{2})}{\Gamma(\frac{d-3}{2})}(\sin\phi)^{d-6} + \ldots + \frac{\sqrt{\pi}\Gamma(\frac{2}{2})}{\Gamma(\frac{3}{2})}\right]$$

$$+ \frac{\Gamma(\frac{d-1}{2})}{\sqrt{\pi}\Gamma(\frac{d}{2})}\phi$$

$$\geq -\frac{1}{d-2}\cos\phi\sin\phi\frac{\Gamma(\frac{d-1}{2})}{\Gamma(\frac{d-2}{2})}\left[\sum_{i=1}^{\lceil\frac{d-4}{2}\rceil}\frac{(\sin^2\phi)^i}{\sqrt{\frac{2i+1}{2}}} + 1\right] + \frac{\Gamma(\frac{d-1}{2})}{\sqrt{\pi}\Gamma(\frac{d}{2})}\phi \quad \text{(by Gautschi's Inequality)}$$

$$\geq -\frac{1}{d-2}\cos\phi\sin\phi\frac{\Gamma(\frac{d-1}{2})}{\Gamma(\frac{d-2}{2})}\left[\sum_{i=1}^{\lceil\frac{d-4}{2}\rceil}\frac{(\sin^2\phi)^i}{\sqrt{\frac{2i}{2}}} + 1\right] + \frac{\Gamma(\frac{d-1}{2})}{\sqrt{\pi}\Gamma(\frac{d}{2})}\phi$$

$$= -\frac{1}{d-2}\cos\phi\sin\phi\frac{\Gamma(\frac{d-1}{2})}{\Gamma(\frac{d-2}{2})}\left[\sum_{i=1}^{\lceil\frac{d-4}{2}\rceil}\frac{1}{\sqrt{i}} + 1\right] + \frac{\Gamma(\frac{d-1}{2})}{\sqrt{\pi}\Gamma(\frac{d}{2})}\phi$$

$$\geq -\frac{1}{d-2}\cos\phi\sin\phi\frac{\Gamma(\frac{d-1}{2})}{\Gamma(\frac{d-2}{2})}\left[2\sqrt{\frac{d-4}{2}} + 1\right] + \frac{\Gamma(\frac{d-1}{2})}{\sqrt{\pi}\Gamma(\frac{d}{2})}\phi.$$

Since $\phi = \cos^{-1}\left(\frac{1+\epsilon}{2\|w^*\|}\right)$, then

$$K \geq -\frac{1}{(d-2)}\frac{1+\epsilon}{2\|w^*\|}\sqrt{1 - \frac{(1+\epsilon)^2}{4\|w^*\|^2}}\frac{\Gamma(\frac{d-1}{2})}{\Gamma(\frac{d-2}{2})}\left[2\sqrt{\frac{d-4}{2}} + 1\right] + \frac{\Gamma(\frac{d-1}{2})}{\sqrt{\pi}\Gamma(\frac{d}{2})}\cos^{-1}\left(\frac{1+\epsilon}{2\|w^*\|}\right).$$

Again by Gautschi's Inequality we obtain:

$$\frac{\Gamma(\frac{d-1}{2})}{\Gamma(\frac{d-2}{2})} < \sqrt{\frac{d-1}{2}},$$

and hence,

$$K > -\frac{1}{(d-2)}\frac{1+\epsilon}{2\|w^*\|}\sqrt{1 - \frac{(1+\epsilon)^2}{4\|w^*\|^2}}\sqrt{\frac{d-1}{2}}\left[2\sqrt{\frac{d-4}{2}} + 1\right] + \frac{\Gamma(\frac{d-1}{2})}{\sqrt{\pi}\Gamma(\frac{d}{2})}\cos^{-1}\left(\frac{1+\epsilon}{2\|w^*\|}\right).$$

Thus, we have that:

$$\frac{\Gamma\left(\frac{d}{2}\right)}{\sqrt{\pi}\Gamma\left(\frac{d-1}{2}\right)}\int_0^\phi \sin^{d-2}\theta d\theta > -\sqrt{\frac{d}{2\pi}}\frac{1}{(d-2)}\frac{1+\epsilon}{2\|w^*\|}\sqrt{1 - \frac{(1+\epsilon)^2}{4\|w^*\|^2}}\sqrt{\frac{d-1}{2}}\left[2\sqrt{\frac{d-4}{2}} + 1\right]$$

$$+ \frac{1}{\pi}\cos^{-1}\left(\frac{1+\epsilon}{2\|w^*\|}\right).$$

Hence, assuming $\|w^*\| = \frac{\sqrt{d}}{c}$, we obtain:

$$\lim_{d\to\infty}\frac{\Gamma\left(\frac{d}{2}\right)}{\sqrt{\pi}\Gamma\left(\frac{d-1}{2}\right)}\int_0^\phi \sin^{d-2}\theta d\theta \geq -\frac{c(1+\epsilon)}{2\sqrt{2\pi}} + \frac{1}{\pi}\frac{\pi}{2} = \frac{1}{2} - \frac{c(1+\epsilon)}{2\sqrt{2\pi}}.$$

Note that we have:

$$\mathbb{P}_{w^{(0)}}\left(\langle w^{(0)}, \frac{w^*}{\|w^*\|}\rangle \geq \frac{(1+\epsilon)}{2\|w^*\|}\right) \leq \mathbb{P}_{w^{(0)}}\left(\langle w^{(0)}, \frac{w^*}{\|w^*\|}\rangle \geq 0\right) = \frac{1}{2},$$

and hence, we conclude that:

$$\frac{1}{2} - \frac{c(1+\epsilon)}{2\sqrt{2\pi}} \leq \lim_{d \to \infty} \frac{\Gamma\left(\frac{d}{2}\right)}{\sqrt{\pi}\Gamma\left(\frac{d-1}{2}\right)} \int_0^\phi \sin^{d-2}\theta d\theta \leq \frac{1}{2}.$$

$\square$

