# OpenReview forum: "LLBoost: Last Layer Perturbation to Boost Pre-trained Neural Networks"
_ICLR.cc/2021/Conference — Reject_

### Official Review · AnonReviewer2 · 2020-10-26
**Adjusting the last linear layer without changing the training accuracy is interesting, but the over-parametrized assumption rarely holds in practice**

**Rating:** 5
**Confidence:** 2

**Review:**

##########################################################################

Summary:

This paper proposes LLBoost that enables adjusting the last linear layer without impacting the training accuracy under the assumption that the last linear layer is in an over-parametrized situation. When the last layer is not over-parametrized, LLBoost first applies the low rank approximation to the training feature matrix through the SVD decomposition, which may affect the original training accuracy. The reason why LLBoost does not change the training accuracy is explained as follows: In an over-parametrized noiseless linear regression, a solution of a linear system $y = wX$ obtained by the gradient descent with an initial value of $w^{(0)}$ is given in a closed form of $\hat{w} = w^{(0)} (I - X(X^\top X)^\dagger X^\top) +  yX^\dagger$. Therefore, we can compute a solution of $y = wX$ by simply generating $w^{(0)}$ randomly and applying this formula. It is also experimentally verified that LLBoost can adjust the last linear layer without impacting the training accuracy (after appriximated with SVD when necessary). The authors also present theoretical results that sampling $w^{(0)}$ uniformly on the hyper-shpere of appropriate radius leads to a solution that is better than the minimum norm solution ($yX^\dagger$) with constant probability.

##########################################################################

Reasons for score:

Overall, I vote for weak reject. It is interesting that LLBoost can adjust the last layer without impacting the training accuracy. And the theoretical results give a reason to sample $w^{(0)}$ uniformly on a hyper-sphere in Alg. 1. However, the condition that the last layer is over-parametrized is rarely satisfied in practical problems to which DNNs are applied. As discussed in Sec. 4, the low rank approximation can harm the accuracy in large problems like ImageNet.
The authors show in Figure 1/2/3,  that LLBoost can improve the validation accuracy without impacting the training accuracy. However, since Alg. 1 directly uses the validation labels (though it is denoted by 'test labels' in Alg. 1) to select $w_{best}$, it should be compared in terms of the 'hold-out' test accuracy to examine the usefulness of LLBoost.


##########################################################################

Pros:

1. The authors propose a method to adjust the last linear layer of a DNN without impacting the training accuracy, under the assumption that the last layer is over-parametrized.

2. The authors give theoretical results that sampling $w^{(0)}$ on a hyper-sphere leads to a good solution with constant probability.


##########################################################################

Cons:

1. In practical problems to which DNNs are applied, the over-parametrized assumption rarely holds. And the low rank approximation with SVD may worsen the accuracy.

2. Because Alg. 1 uses validation labels as its input to select the best solution, it is not enough to report the validation accuracy in experimental results like Figure 1/2/3. The accuracy of a 'hold-out' test set should also be reported.

##########################################################################

Other concerns:

- In line 3 of Alg. 1, why $P=I_{d \times d} - U I_{r \times r} U^\top$ is used, though it is explaned as $P=I -  X(X^\top X)^\dagger X^\top$ in Method Overview paragraph.

- Although the input of Alg. 1 includes 'test feature matrix' and 'test labels', they seem better denoted by 'validation feature matrix' and 'validation labels', respectively.

- In Figure 2, Train/Val. Acc. (Original) should be values without low rank appriximations (e.g. 95.193% for train acc. in ImageNet as denoted in Sec. 4). Also, it is not clear whether the val. acc. is computed with or without low rank appriximations.

---

> ### Author Response · Authors · 2020-11-18
> **Response to Reviewer 2**
>
> Thank you for your review.  We address your concerns in detail below.
>
> * “In practical problems to which DNNs are applied, the over-parametrized assumption rarely holds. And the low rank approximation with SVD may worsen the accuracy.”
>     * Recently, the over-parameterized setting is a topic of major interest in deep neural networks since most neural networks are over-parameterized in practice (see https://arxiv.org/abs/1611.03530, https://arxiv.org/abs/1912.02292), and on small datasets that require transfer learning, neural networks are again almost always over-parameterized.  In the related works section of our paper, we have provided several examples of works (such as https://arxiv.org/abs/1812.11118) that highlight the benefits of overparameterization in modern machine learning.  This implies that our LLBoost results are relevant to a broad setting.
>     * The low rank approximation will only decrease the training accuracy for under-parameterized networks (such as those trained on full ImageNet).  For example, our approximation did not impact the training accuracy of pre-trained ResNets on CIFAR10, which are over-parameterized.
> * “Because Alg. 1 uses validation labels as its input to select the best solution, it is not enough to report the validation accuracy in experimental results like Figure 1/2/3. The accuracy of a 'hold-out' test set should also be reported.”
>     *  In Appendix G (Figure 11) and as indicated by Reviewer 1, we already provide empirical evidence that using LLBoost to improve validation accuracy can also lead to an improvement in test accuracy.  In other words, we show that LLBoost does not overfit the validation set. Theorem 3 of the paper also mathematically proves that LLBoost decreases the test error (not just the validation error) of any interpolating solution with high probability.
> * “In line 3 of Alg. 1, why $P = I_{d \times d} - U I_{r \times r} U^T$ is used, though it is explaned as $P = I - X (X^T X)^{\dagger} X^T$ in Method Overview paragraph.”
>     * The two different formulations of the projection term are exactly the same. $P = I - X(X^TX)^{\dagger}X^T = I - U\Sigma  V^T(V\Sigma^2V^T)^{\dagger} V \Sigma U^T=I-U\Sigma(\Sigma^2)^{\dagger}\Sigma U^T=I-UI_{r\times r}U^T$.  We just use one formulation over the other in our Algorithm since it can be more computationally efficient to compute.
> * “Although the input of Alg. 1 includes 'test feature matrix' and 'test labels', they seem better denoted by 'validation feature matrix' and 'validation labels', respectively.”
>     * Thank you for your note on denoting test feature matrix / test labels as validation feature matrix / validation labels. We are happy to update this in our paper.
> * “In Figure 2, Train/Val. Acc. (Original) should be values without low rank appriximations (e.g. 95.193% for train acc. in ImageNet as denoted in Sec. 4). Also, it is not clear whether the val. acc. is computed with or without low rank appriximations.”
>     * We are happy to update this in the revision.  To clarify, the validation accuracy does not need to be computed with low rank approximations.  The low rank approximation is only needed for the training feature matrix in order to be able to search the space of initializations that are orthogonal to it.

---

> > ### Comment · AnonReviewer2 · 2020-11-20
> > **additional comments regarding the response**
> >
> > In my understanding, LLBoost requires that the last linear layer itself has to be over-parametrized, that is, the size of input to the last layer must be larger than the number of training samples. If this is true, recent topics related to the over-parametrized setting in DNN has less relation to this problem.
> >
> > I recommend to include Figure 11 in the main paper, not in the  appendix. And if possible, it is better to evalate test acc for all settings in Figure 2.

---

> > > ### Author Response · Authors · 2020-11-20
> > > **Follow up**
> > >
> > > Thank you again for the quick follow up.
> > >
> > > * LLBoost just requires that the incoming training feature matrix is either low rank or well-approximated with a low rank matrix.  A sufficient condition for this requirement is that the size of the last layer is larger than the number of samples, but this is not necessary.  In particular, this requirement was always satisfied for over-parameterized neural networks in our experiments (we note that fix-resnext-101 on full ImageNet was not over-parameterized since we could not get 100% training accuracy).  For example, in Figures 9 and 10 (in the attached Supplementary Material), even when using 50,000 examples from CIFAR10, we could use less than 50 components to approximate the training feature matrix without noticing any decrease in training accuracy.  This is additionally indicated in Figure 9 by the rapid drop in the spectrum of the training feature matrix.
> > >
> > > * "I recommend to include Figure 11 in the main paper [...]"
> > >     * Thank you for the suggestion.  We are currently working on running additional experiments to demonstrate the boost in test accuracy.  We would be happy to update the paper once these experiments are complete.   In the meantime, we would again like to emphasize the importance of our theoretical result (Theorem 3), which guarantees that our method boosts test performance.

---

> > > > ### Author Response · Authors · 2020-11-23
> > > > **Paper Updates**
> > > >
> > > > Just to follow up on our previous comment and your suggestions, we have updated our paper with the following:
> > > >
> > > > * We have updated the text in Algorithm 1 to state "validation" instead of "test".
> > > > * We have added an additional Figure 12 to the Appendix, which provides more evidence that LLBoost provides a boost to both validation and test accuracy over pre-trained resnets.  In particular, this figure demonstrates that LLBoost improves transfer learned resnet models on small image classification datasets.  We are still working on evaluating all the test accuracies for the models in Figure 2, but we hope the new figure alleviates your concerns regarding the boost to the test performance.   Thank you again for your suggestions.

---

> > > > > ### Comment · AnonReviewer2 · 2020-11-24
> > > > > **Over-parametrization**
> > > > >
> > > > > Thank you for your comments and updates.
> > > > >
> > > > > I understand that LLBoost can tune the last layer without impacting the training accuracy when the training feature matrix is low rank. (The condition that "the size of the last layer is larger than the number of samples" is a sufficient condition for this, but not necessary.)

---

### Official Review · AnonReviewer3 · 2020-10-26
**An innovative way to improve generalization performance**

**Rating:** 6
**Confidence:** 3

**Review:**

################################################################

Summary:

This paper provided an efficient algorithm (LLBoost) to boost the validation accuracy without spending too much time tuning hyperparameter. The algorithm is theoretically and empirically guaranteed.

################################################################

Reason for Score:

This paper provides an innovative way to improve generalization performance. My major concern is about experiment part. Since the algorithm use valid data to tune the parameter, it should another held-out test data to show the result. However, the author only did experiment on test-data for model ResNet-18, which is not sufficient to support the paper.

################################################################

pros:

1, This paper gave an efficient LLBoost algorithm to quickly improve the validation accuracy. The algorithm is theoretically guaranteed.

2, The paper clearly stated the intuition of the algorithm. The paper considered models that have fc layer as the last layer (most of the current models have this property), and transformed the problem into a linear regression problem.

3, A surprising point of the algorithm is that it does not impact the training loss.

################################################################

cons:
1, While the algorithm has a theoretically guarantee, the experiment part did not convince me. This is the major concern for the paper. The author tune the parameter using valid data and say valid accuracy is improved, which is not enough. It should another held-out test data to show the result for all the experiment. However, the author did an experiment on test-data only for ResNet-18, which is not sufficient to support the paper. Also, The author should put this test-data-ResNet-18 experiment in main part of the paper, not Appendix.

2, Section 3 (preliminaries and method) is not well-organized. I cannot see why the author put this two lemmas here.

(1) why "Lemma 1 implies that LLBoost does not affect training predictions since it only ever adds a component orthogonal to the span of the training feature matrix"? The lemma 1 seems having nothing to do with the LLBoost algorithm

(2) what is the purpose for lemma 2? The paper doesn't clearly state it.

I understand the reason after the author explained it in response. But I strongly suggest the author to explain it in paper for the final version.

3, Based on my understanding of this paper, the algorithm has to be applied to an existing pretrained model which is sufficient good. If we don't have a good pretrained model, does this algorithm provide a better (or comparable) result than the well-tuned model? I am just curious about it and hope the author to do some experiments in the future.

---

> ### Author Response · Authors · 2020-11-18
> **Response to Reviewer 3**
>
> Thank you for your review and the positive comments.  We address your concerns below.
>
> * “If we don't have a good pretrained model, does this algorithm provide a better (or comparable) result than the well-tuned model?”
>     * The purpose of applying LLBoost on pre-trained models was to show that even on the well trained models, LLBoost can provide significant boosts in test accuracy.  This does not imply that a practitioner would need a strong model to apply LLBoost.  While testing the method, we found LLBoost continued to improve performance even on the non-pretrained models.  We are happy to show empirical evidence for this if desired.
> * “why "Lemma 1 implies that LLBoost does not affect training predictions since it only ever adds a component orthogonal to the span of the training feature matrix"? The lemma 1 seems having nothing to do with the LLBoost algorithm”
>     * Lemma 1 is important to include in Section 3 as it theoretically explains why LLBoost does not affect training accuracy. Since $w^{(0)}(I - X(X^TX)^{\dagger} X^T)$ is orthogonal to the training feature matrix, when multiplying the training feature matrix $X$ by the new weight term $w=w^{(0)}(I - X(X^TX)^{\dagger} X^T)+yX^{\dagger}$, $w^{(0)}(I - X(X^TX)^{\dagger} X^T)$ will just equal  0. As a result, adding $w^{(0)}(I - X(X^TX)^{\dagger} X^T)$ to the pseudo-inverse solution neither impacts the training accuracy nor the training predictions.
> * “what is the purpose for lemma 2?”
>     * Lemma 2 presents LLBoost’s method for sampling uniformly on the unit sphere (lines 5-6 of Algorithm 1). In other words, Lemma 2 states that if you sample $z \sim \mathcal{N}(0, I_{d \times d})$ and divide $z$ by its norm, then the resulting term is a uniform sample on the unit sphere. Thus, if we desire to sample uniformly on a hyper-sphere of radius $\gamma$, we can easily do so by multiplying $\frac{z}{||z||}$ with $\gamma$.

---

> > ### Author Response · Authors · 2020-11-25
> > **Update**
> >
> > Unfortunately, we were not notified of your change in score and your new listed cons regarding test instead of validation accuracy (there was no new email sent out for the edit, but only for new comments).  In line with our response to other reviewers, we would importantly like to point out that we actually had train/val/test splits in Figure  11 of our original submission and have added even further experiments demonstrating the benefit to both validation and test accuracy in the supplementary (Figure 12).  Additionally, we would like to emphasize that our Theorem 3 actually proves this benefit for test data as well.
> >
> > Please let us know if you have any other concerns.  Thank you again.  If this was the main reason for lowering the score, we hope that in light of the fact that we already had these experiments in our original submission, you would consider re-raising the score.

---

### Official Review · AnonReviewer1 · 2020-10-28
**While having some theoretical justification the method fails to deliver in practice.**

**Rating:** 4
**Confidence:** 4

**Review:**

The paper studies the problem of boosting test performance of the last layer by crafting random perturbations that are orthogonal to the train feature matrix of the last layer (at least in the overparametrized case) thus leaving train performance unaffected.

==========================================

Main Comment:
While the idea seemed interesting to me at first I find the paper overselling the results. The claim that LLboost improves performance is very strong when looking at the numbers: 0.2% improvement for cifar10 and 0.08% improvement for imagenet is marginal to say the least. This is even starker when looking at test performance as opposed to validation where for cifar the improvement is 0.03%-0.1%. The benefit for 2 class imagenet32 is more significant at the low sample regime but also fails to impress, and feels more like cherry picking rather than a serious experimental ablation.

Minor Comments:
Figure 2 (should be a table environment, btw), shows that standard normal perturbations of the last layer reduce accuracies significantly. I'm wondering what would happen if we apply the same variance as the LLboost, as having different variances gives  a false impression of whether random perturbations help or hurt.

---

> ### Author Response · Authors · 2020-11-18
> **Response to Reviewer 1**
>
> Thank you for your review. We address your concerns below.
>
> * “While the idea seemed interesting to me at first I find the paper overselling the results.”
>     * We would like to point out that all of the models in our experiments were pre-trained or state-of-the-art, meaning that producing any improvement at all is inherently difficult. Traditionally, improving such state-of-the-art models even requires training on random seeds, a computationally expensive task that our method does not require. In addition, we have a mathematical proof that guarantees a boost in test accuracy for over parameterized linear models and for over-parameterized neural networks where the last layer is linear.
> * “The benefit for 2 class imagenet32 is more significant at the low sample regime but also fails to impress, and feels more like cherry picking rather than a serious experimental ablation.”
>     * We respectfully disagree that our experiments are cherry picked. We have intentionally demonstrated the benefit of our model across CIFAR10, full ImageNet, and 7 different subsets of ImageNet-32 using 4 different pre-trained/state-of-the-art models.  As you point out, the benefit of our method in boosting pre-trained models in the low sample regime is significant.  Transfer learning from a pre-trained model is especially useful on small datasets, but improving the performance of a transfer learned model on a small dataset is difficult due to lack of data.  The fact that LLBoost can provide such improvements on a pre-trained model with as little as 100 samples of ImageNet’s classes 1 and 2 has important implications for practitioners.
> * “I'm wondering what would happen if we apply the same variance as the LLboost, as having different variances gives a false impression of whether random perturbations help or hurt.”
>     * Without incorporating the projection operator (as in LLBoost), there is first no guarantee that training predictions are unchanged, and thus, LLBoost already has an advantage over a simple perturbation regardless of the variance.  Moreover, in high dimensions, vectors sampled from an isotropic Gaussian distribution lie on a sphere of radius sqrt(d) with high probability and so reducing the variance would simply reduce the size of this radius.  Now, LLBoost samples its initializations on the unit sphere before multiplying by $\gamma$. Therefore, multiplying the projection operator with a vector with i.i.d Gaussian entries of smaller variances is really just a proxy for LLBoost.  The benefit of LLBoost is that we prove in Theorem 3 that a radius of size, $\gamma$ roughly  $\frac{||w^*||}{\sqrt{d}}$, has a high probability of improving the pseudo-inverrse solution.

---

> > ### Author Response · Authors · 2020-11-23
> > **Further updates to our paper**
> >
> > We have now added additional experiments (Figure 12) to the supplementary to demonstrate that our method provides a consistent improvement to validation and test accuracy for transfer learned models on small datasets.  We would again like to emphasize that improving a transfer learned model without affecting training accuracy is nontrivial the small dataset regime, and that our method is a computationally efficient means of improving models even in this setting.

---

> > ### Comment · AnonReviewer1 · 2020-11-25
> > **Thank you for the response.**
> >
> > I would like to thank the authors for the response.
> > * I agree that it is hard to improve state-of-the-art models, but I'm not convinced that you actually improved the models. The improvement on the test for imagenet and cifar is too insignificant to tell. I think that your comment 'Traditionally, improving such state-of-the-art models even requires training on random seeds' is misleading. The real way to improve state-of-the-art models, is coming up with shrewd algorithms and architectures that improve on previous methods and perhaps trying more than one random seed is part of what some papers are doing, this is in no way the 'right way'.
> >
> > * While I agree that 'cherry-picked' might be too strong, demonstrating 7 different subsets of a more niche dataset that you had success as opposed to two lines of standard benchmarks 'over-representing' the good results vs the bad ones.
> >
> > * Yes, the experiment I'm suggesting is without using the projection operator. It might increase the train error slightly, but if this increases test accuracy, then I don't see why we should mind.
> >
> > =============
> >
> > After taking into account the authors' comments I decided to keep my score unchanged.

---

### Decision · Program_Chairs · 2021-01-07
**Final Decision**

**Decision:**

Reject

**Comment:**

Though the method suggested in this paper is interesting, theoretically motivated, and resulted in some practical improvement, the reviewers ultimately had low scores. The reasons for this are:
1) The improvements obtained by this method were rather small, especially on the standard datasets (CIFAR, Imagenet).
2) In the main results presented in the paper, it seems that a proper validation/test split was not done (which seems quite important for demonstrating the validity of this method). In some of the results, presented in supplementary, such a split was done, but this seems to decrease the performance of the method even more.
3) The method requires that features in the last hidden layer approximately span a low dimensional manifold. This seems like a major limitation for the accuracy of this method, which becomes approximate in datasets where the number of datapoints is larger than the size of the last hidden layer (which is the common case).

Therefore, I suggest the authors try to improve all of the above issues and re-submit. For example, one simple way to address issue 3 and potentially improve the results (issue 1) is to use the same method on all the features in all the layers, instead of just the last layer. In other words, concatenate all the features and all the layers, and then add a linear layer from this concatenated feature vector directly to the network output, in a direction that is orthogonal to the data.